# Climate change has increased the odds of extreme regional forest fire years globally

John T. Abatzoglou [1] ✉, Crystal A. Kolden [1], Alison C. Cullen[2], Mojtaba Sadegh [3], Emily L. Williams[4], Marco Turco [5] & Matthew W. Jones [6]

Regions across the globe have experienced devastating fire years in the past decade with far-reaching impacts. Here, we examine the role of antecedent and concurrent climate variability in enabling extreme regional fire years across global forests. These extreme years commonly coincided with extreme (1-in-15-year) fire weather indices (FWI) and featured a four and five-fold increase in the number of large fires and fire carbon emissions, respectively, compared with non-extreme years. Years with such extreme FWI metrics are 88-152% more likely across global forested lands under a contemporary (2011–2040) climate compared to a quasi-preindustrial (1851–1900) climate, with the most pronounced increased risk in temperate and Amazonian forests. Our results show that human-caused climate change is raising the odds of extreme climate-driven fire years across forested regions of the globe, necessitating proactive measures to mitigate risks and adapt to extreme fire years.

The impacts of fires on society and the environment have received growing scientific and public attention over the past decade. This increased recognition stems not only from the complexity of fire in the Earth system[1,2] but also from the occurrence of extreme fire years—categorized as years with widespread regional fire activity and exceptional burned area. Such extreme fire years include 2019–2020 in Australia[3], 2020 in the western US[4], as well as 2023 in Canada[5]. While individual fires—such as those associated with widespread losses—have garnered immediate media attention[6], extreme fire years have broader implications. These extreme and often prolonged and widespread fire years impact air quality[1], elevate carbon emissions[7], overextend social support infrastructure[8] and challenge international coordination of fire suppression resource[9].

While global burned area has declined in recent decades due to land-use change in fire-frequent savannas[10], burned area and fire carbon emissions have increased in the world's forested biomes[11,12]. Increases in fire activity have been pronounced in places where increased fire weather[13] and longer fire weather seasons[14] coincide with fuel-rich, flammability-limited lands such as the western United States, Canada, and Siberia[15,16]. Both longer and more intense periods of fire weather

have facilitated increased geographic synchronization in fire weather, which can overwhelm fire suppression capabilities[17,18]. Complementary to changing biophysical factors that influence fire potential, local-to-regional anthropogenic actions have affected trajectories of fire activity. The legacy effects of a century of fire suppression and agricultural land abandonment have increased fuel loads and fuel continuity in some regions that predispose landscapes to larger, more energetic fires[19]. Contemporary fire suppression also largely constrains burned area in some regions to primarily occurring under extreme fire weather conditions—augmenting the sensitivity of fire to climate drivers[20]. In tropical forests, deforestation occurs to make space for globalized industrial agriculture—resulting in an economic catalyst to regional fire activity that becomes more prominent during drought years[21].

A growing body of attribution research has examined the role of anthropogenic climate change on[22,23] fire activity, whether measured in burned area[22–24], through extreme fire typologies[25], or specific extreme fire events[26–28]. Such research generally isolates the contributions of climate factors, even though fire responses (e.g., burned area) depend on a complex interplay of social, biological, and physical drivers that may confound direct climate attribution[29]. These studies typically, but

[1]School of Engineering, University of California, Merced, CA, USA. [2]Evans School of Public Policy and Governance, University of Washington, Seattle, WA, USA. [3]Department of Civil Engineering, Boise State University, Boise, ID, USA. [4]Sierra Nevada Research Institute, University of California, Merced, CA, USA. [5]Department of Physics, University of Murcia, Murica, Spain. [6]Tyndall Centre for Climate Change Research, School of Environmental Sciences, University of East Anglia, Norwich, UK. ✉e-mail: jabatzoglou@ucmerced.edu

**Table 1 | Contributing factors, sources, and selected impacts for extreme fire years—defined as years with the highest annual burned area during 2002–2023 for case study regions**

| Case study region (year) & *Forest biome(s)* | Contributing factors to fire year | Key impacts | Anomalies (% departure of average for non-extreme years) | | | |
|---|---|---|---|---|---|---|
| | | | Burn Area | Fire Count | Very Large Fires | Fire Emissions |
| SE Asia (2007) *Tropical* | • moderate drought[69]<br>• slash-and-burn for palm oil[70] | • high levels of smoke/haze[70]<br>• high methane emissions from peatland[71] | 224* | 136+ | 176 | 316* |
| Amazon (2010) *Tropical* | • low precipitation<br>• deforestation[72] | • significant global carbon emissions | 127* | 1 | 263* | 136* |
| Siberia (2012) *Boreal* | • severe drought | • air quality impacts both locally and downwind[73] | 143* | 100+ | 95 | 116* |
| Mediterranean (2017) *Temp Conifer/ Broadleaf* | • drought + heatwave[40]<br>• agricultural land abandonment | • 60+ deaths, 100+ injuries[74]<br>• 2000+ homes destroyed | 63* | 63 | 620* | 207* |
| SE Australia (2019–2020) *Temp Broadleaf* | • drought[75]<br>• strong wind events[76]<br>• logging in forest plantations[77] | • widespread wildlife mortality<br>• 33 direct deaths 400+ smoke-related premature deaths, loss of 3000+ houses[78] | 1895* | 163* | 1205* | 2180* |
| Western US (2020) *Temp Conifer* | • drought[79]<br>• downslope winds[43]<br>• Fuel accumulation[80] | • $20 billion impact<br>• 40 deaths[81] | 323* | 0 | 311* | 490 |
| Chile (2023) *Temp Conifer/ Broadleaf/* | • severe drought, downslope winds[82]<br>• tree plantations[83] | • 1554 homes destroyed<br>• 26 deaths | 1162* | 37 | 1067+ | 525* |
| Canada (2023) *Boreal/ Temp Conifer* | • record temperature, persistent blocking[5]<br>• excess fuel from fire suppression[84] | • ¼ of global annual tree loss[85]<br>• 230,000+ people evacuated[5] | 642* | 290* | 538* | 1190* |

The three right columns show anomalies (percent departure from the average of all non-extreme fire years, expressed as a percent departure from the average) in burned area, the number of fires, number of fires with burned area greater, or equal to 100 km², and fire carbon emissions. Notations of * and + correspond with years with the highest and second-highest values, respectively, during 2002–2023.

not always[26], show that human-induced climate change has augmented the likelihood or magnitude of fire weather extremes and fire potential. Yet, most of these efforts have been constrained geographically to specific case study regions and used different methods, limiting the ability to quantitatively compare the role that human-caused climate change has played in fires across geographies. Conversely, at the global scale, most prior climate-fire analyses have focused on correlative assessments of intra-to-interannual variability[2,16], or on individual extreme fire events occurring on the scale of weeks to months[30,31], but not extreme fire years.

Here, we analyze the role of fire weather in enabling extreme fire years across forested ecoregions of the world and quantify the contribution of anthropogenic climate change to those extreme years. We first identify commonalities in both fire weather indices and antecedent moisture in these extreme regional fire years using representative case study regions across major forest biomes[32] alongside a systematic analysis of forested ecoregions worldwide. Our analysis is limited to predominantly forest and woodland areas, where fire produces significant carbon and smoke and where climate-fire relationships are more clearly coupled due to the absence of fuel limitations[16,32]. While a singular definition of extreme in fire science is absent[31,33,34], herein we adopt the term "extreme fire years" as the year with the greatest amount of burned area in MODIS record (2002–2023) for a given region. Leveraging this empirical analysis of observed extreme regional fire years, we systematically apply an attribution framework to quantify how anthropogenic climate change has altered the probability of climate-enabled extreme fire years in forested regions globally.

## Results

### Conditions underpinning extreme fire years in case studies

Extreme regional fire years generally coincided with exceptional or record high fire weather, quantified by the Fire Weather Index (FWI, calculated from the Canadian Forest Fire Danger Rating System) for our case study regions (Table 1; Fig. 1c). Notably, FWI metrics including the number of days per year above the 95th percentile (FWI$_{95d}$), the maximum 90-day moving average FWI (FWI$_{fs}$), and the annual maximum FWI (FWI$_{max}$) were at their highest values during the 1979–2023 period for extreme fire years in the western US in 2020, Canada in 2023, SE Australia in 2019–20, Chile in 2022–2023, and Siberia in 2012. All case study regions except the Amazon and SE Asia regions had FWI metrics exceeding the 1-in-15 return period during the record fire year —codified with a generalized extreme value (GEV) approach using data from a 1979–2023 baseline—suggestive of the role of climate variability in enabling and driving extreme fire years (Figs. S1 and S3). While extreme fire years in the Amazon and SE Asia regions occurred with moderate FWI and drought, these years were strongly influenced by deforestation and land clearing fires—which occurred earlier in the study period and have waned over the past couple decades (Fig. 1b; Table 1).

Interannual relationships between forested burned area and FWI metrics showed moderate-to-strong positive correlations, suggesting flammability-limited fire regimes (Figs. S1–S3). By contrast, variation in moisture availability (precipitation or evapotranspiration) in the previous 1–2 years did not correlate significantly with forested burned area (Figs. S4 and S5). Moisture availability was not particularly anomalous prior to the most extreme fire years (Fig. 1c), except in Chile, where very low precipitation was seen a year prior to the most extreme fire year.

The total number of fires and the number of very large fires (≥100 km²) were anomalously high during extreme fire years in these case study regions. Most regions had at least three times as many very large fires during extreme fire years as the average of other years (Table 1), with six of the eight regions having the most or second most on record. Likewise, extreme fire years typically coincided with the years with the highest fire carbon emissions (e.g., carbon emissions during the 2019–20 fire year in SE Australia were comparable to the

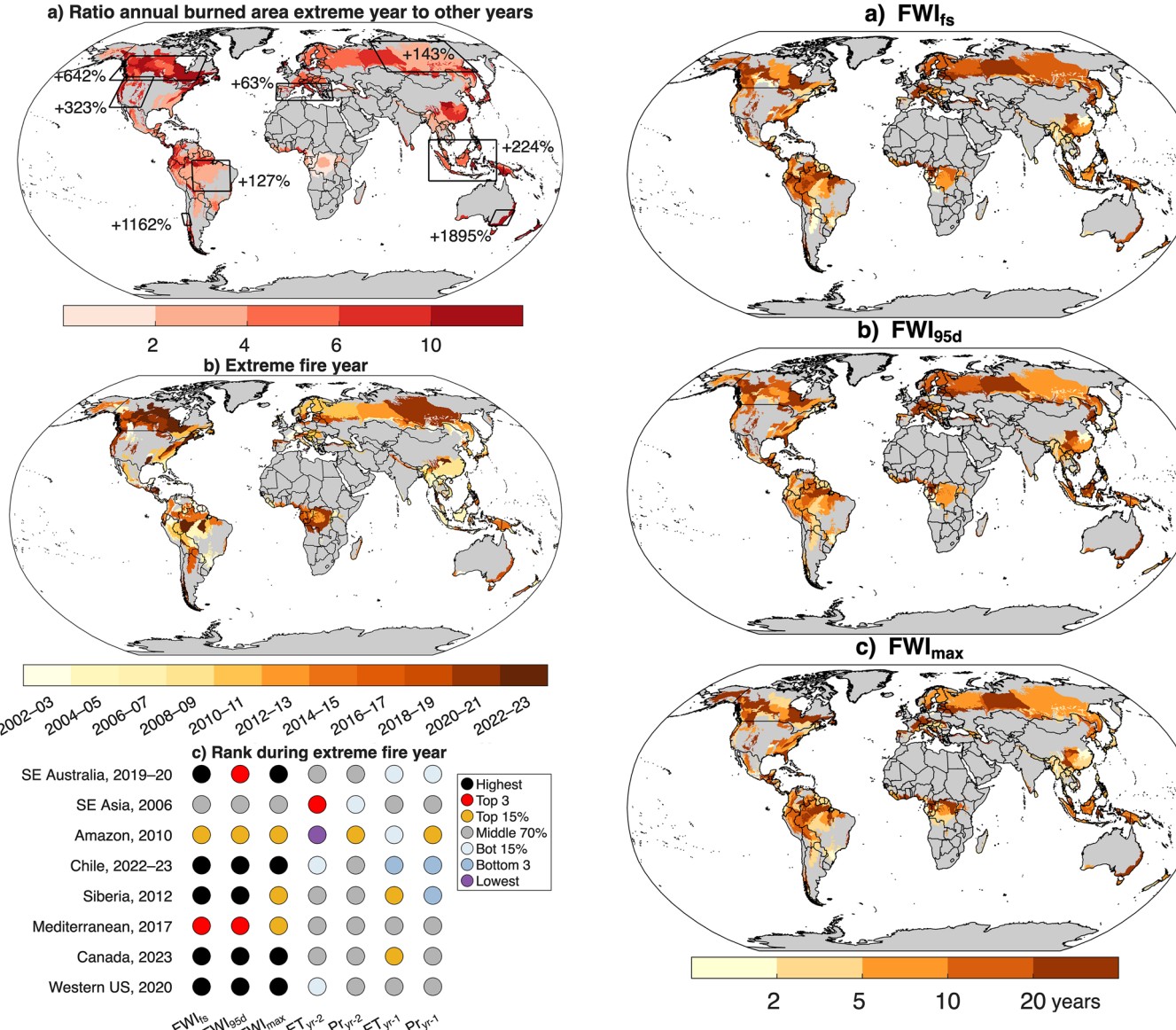

**Fig. 1 | Extreme regional fire years across forested areas during 2002–2023.**
**a** The ratio of burned area during the largest year during 2002–2023 to the mean burned area from all other years. Non-forested ecoregions are masked out. Black boxes show eight case study areas highlighted in the literature (Table 1). **b** Year with the largest forested burned area during 2002–2023 at the ecoregion scale.
**c** Quantitative ranking relative to 1979–2023 of three Fire Weather Index (FWI) metrics (FWIfs: max 90 d moving mean FWI during the year; FWI95: number of days of 95th percentile FWI, FWImax: maximum daily FWI in the year) coincident to, and $ET_{yr-2}$ (evapotranspiration) and $Pr_{yr-2}$ (precipitation) from two years and $ET_{yr-1}$ and $Pr_{yr-1}$ one year prior to the year with the greatest burned area during 2002–2023.

**Fig. 2 | Extreme regional fire years co-occur with extreme fire weather.** Estimates of the return period of Fire Weather Index (FWI) metrics by ecoregion for (**a**) $FWI_{fs}$ (max 90 d moving mean FWI) (**b**) $FWI_{95d}$ (number of days of 95th percentile FWI), and (**c**) $FWI_{max}$ (annual maximum FWI) corresponding with the largest fire year during the 2002–2023 period. Return periods are calculated with respect to 1979–2023. Ecoregions with <20% forest land are excluded and shaded gray.

sum of all remaining years during 2002–2023). The impacts from these extreme regional fire years included widespread hazardous air quality, direct toll on human life and infrastructure, economic losses, and widespread ecosystem impacts (Table 1).

## Extreme fire years across forested ecoregions typically coincide with extreme fire weather

Across forested ecoregions globally, extreme fire years generally coincided with high-to-record FWI metrics, like our case studies (Fig. 2). Extreme regional fire years often occurred in years with extreme FWI (ecoregion median GEV found 1-in-15-years for FWIfs,

1-in-12-years for FWI95d and 1-in-9-years for FWImax). Extreme fire years coinciding with more moderate FWI years were typically located in tropical regions subject to deforestation impacts. By contrast, return intervals for anomalously high precipitation and evapotranspiration from the prior one to two years exhibited comparatively little signal (Fig. S6a, b, e, f), suggestive of negligible generalizable influence of increased fuel load enabling extreme fire years. Likewise, GEV analyses of annual precipitation and evapotranspiration during the prior one to two years show no commonality of extreme longer-term drought beyond the fire year in facilitating extreme regional fire years (Fig. S6c, d, g, h). We adopted the 1-in-15-year FWI in our attribution exercise, given that extreme fire years in six of the eight case studies and a third of forested global ecoregions exceeded this return interval for all FWI metrics.

In most ecoregions, forested burned area during the most extreme fire year was at least five times larger than the average for non-

## a) Number of Fires

## b) Number of Very Large Fires

## c) Fire carbon emissions

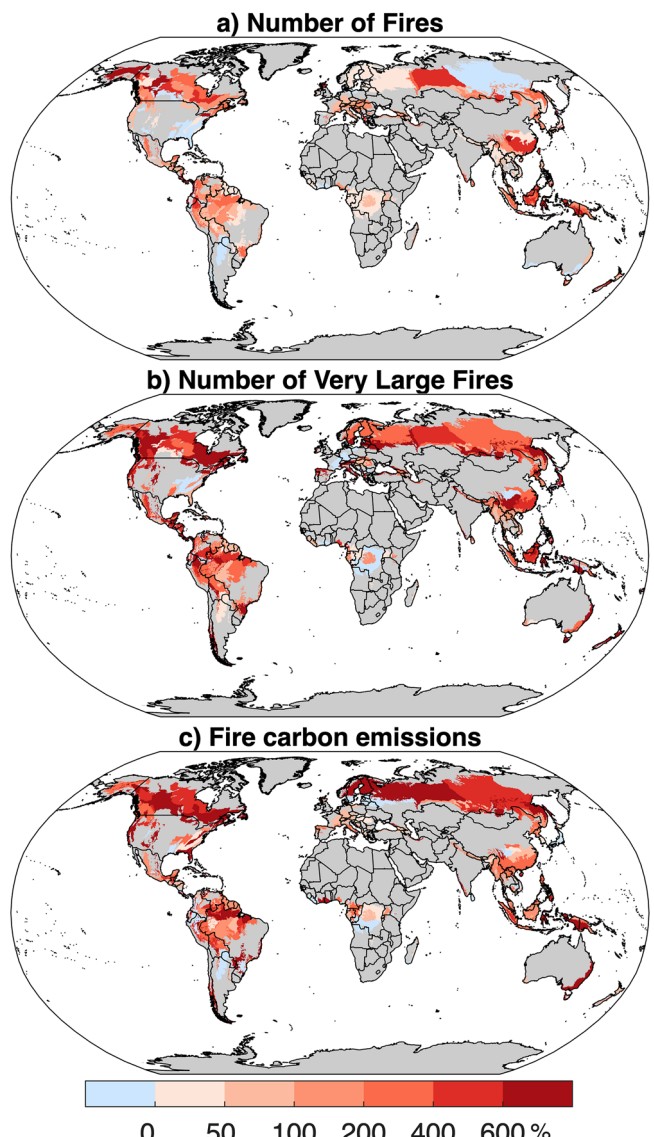

**Fig. 3 | Fire characteristics during extreme regional fire years.** Anomaly (% departure) in (**a**) number of fires, (**b**) number of very large fires, and (**c**) fire carbon emissions from forest and deforestation during the fire year with the highest forest burned area during the study period, compared to the average annual values during all other years. Very large fires are defined as the top 1% of fires by size for each forested ecoregion during the period of record. Ecoregions with <20% forest land are excluded and shaded gray.

extreme years (Fig. 1a). Like results for the case studies, extreme fire years at ecoregion scales were associated with increased fire occurrence (73% median increase across ecoregions) with more substantial increases in very large fires (321% median increase) relative to non-extreme fire years (Fig. 3a, b). Likewise, carbon emissions from forest and deforestation sources were many times higher (405% median increase) than in non-extreme fire years (Fig. 3c).

### Climate change has increased the odds of extreme regional forest fire years across the globe

Climate model results show that climate change has increased the probability of 1-in-15-year FWI extremes that have been conducive to extreme regional fire years (Fig. 4). Many regions show risk ratios (RR) exceeding two under a contemporary (2011–2040) climate relative to that of a quasi-preindustrial (1851–1900) climate. The RR is calculated as the ratio of the probability of FWI extremes

exceeding the benchmark 1-in-15-year value under the contemporary climate to the counterfactual (here quasi-preindustrial) climate. We additionally find broad model agreement of large increases in the Western US, Mediterranean, and Amazon regions, where nearly all models show RR ≥ 2 for $FWI_{fs}$ and $FWI_{95d}$. Only Canada and Siberia regions had a few models with no change or a reduction in probability of FWI extremes, corresponding with increased summer precipitation (Fig. S7). Lastly, no systematic differences in RR across climate models were seen as a function of climate sensitivity or the increase in global mean temperature through 2040 (Fig. S7).

More generally across global forest ecoregions, the median RR of extreme FWI metrics was 1.88–2.52 across metrics (multi-model median across the three FWI metrics), suggesting much higher likelihoods of such extremes under a contemporary climate than under quasi preindustrial climate (Fig. 5; Table S1; Tables S2–S4 for individual models). The largest RR was seen in Amazonian ecoregions, where the likelihood of FWI extremes increased by over fourfold. A majority of tropical and temperate ecoregions showed strong intermodel agreement of increased RR (Fig. S8). Risk ratios were more uncertain across models in magnitude, and in some cases in the direction of change, in the boreal forests of northwestern Canada, Alaska, and Siberia. Overall, we show a doubling in the likelihood of 1-in-15-year FWI metrics (RR ≥ 2) for 46–65% (multi-model median change) of forested areas globally under contemporary climate compared to quasi-preindustrial climate, and a decrease in the likelihoods of extremes (RR ≤ 1) for 3–8% of forested area (Table S1). Results for individual models show widespread agreement of increased RR (Tables S2–S4). The more muted change in RR shown here compared to case study regions is likely a product of the smaller spatial scales of ecoregions that weaken signal-to-noise ratios[35]. The same factors somewhat limit a fair comparison of RR across ecoregions due to the variety in their spatial scales.

The largest increase in RR was seen for $FWI_{fs}$ and $FWI_{95d}$, with the lowest change for $FWI_{max}$ (Table S1). The more limited increase in $FWI_{max}$ is consistent with the weaker emergence of this metric with future climate change[36]. Many of the most extreme fire weather days realized through $FWI_{max}$ are associated with wind extremes concurrent with dry fuels, the former of which exhibits more uncertainty to anthropogenic climate forcing[37]. By contrast, widespread increases in temperature and declines in surface relative humidity facilitate drier fuels (as realized through build-up indices) that drive robust increases in $FWI_{fs}$ and $FWI_{95d}$[13].

## Discussion

Prior research has shown strong intra-to-interannual climate-fire relationships at regional scales[2,15,16] and the role of weather in driving extreme fire events[30]. This study adds to this body of work by demonstrating that extreme regional fire years across forested areas of the globe typically coincide with statistically rare fire weather conditions (e.g., chronically high FWI, days exceeding the 95th percentile FWI), quantified with a 1-in-15-year recurrence interval. While previous studies have alluded to such relationships on limited geographic scales in the modern and historical record[4,38–40], our findings show these results to hold for most forested lands globally outside of regions dominated by deforestation and forest degradation. Similar to other climate extremes, the FWI extremes examined here are a product of multiple interacting drivers on the weather-climate continuum. These drivers include prolonged periods of simultaneous warmth and aridity[41], accentuated by persistent meteorological patterns[42], and capstone weather events at the apex of fuel drying, such as dry frontal passages, pyrocumulonimbus events, and downslope winds[43].

Extreme regional fire years in temperate and boreal forests, which tended to occur in the latter portion of the record, often occurred in years with extreme FWI metrics. Extreme fire years in temperate forests of the Mediterranean Basin, Chile, the western US, and Australia have been additionally exacerbated by excess fuel abundance and fuel

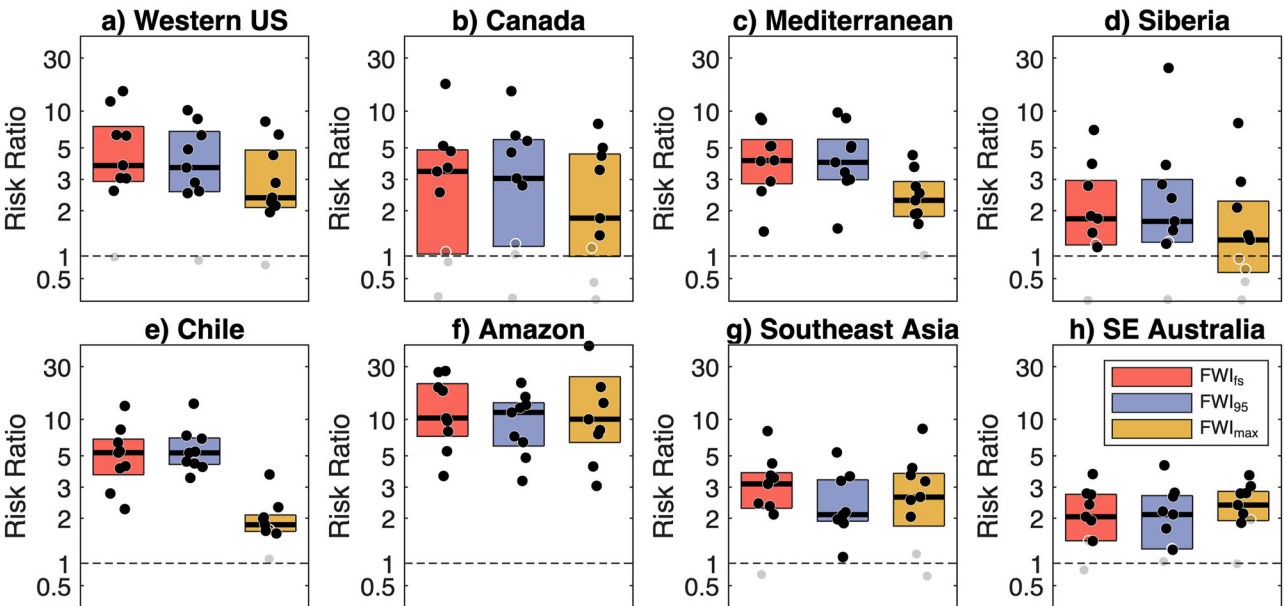

**Fig. 4 | Increased odds of extreme fire weather years due to climate change.** Risk ratio of Fire Weather Index (FWI) extremes between contemporary (2011–2040) and quasi preindustrial (1851–1900) climates for the **a**–**h** eight case study regions. The boundaries of each case study region are outlined in Fig. 1. Box plots are shown separately for each FWI metric (FWI$_{fs}$–maximum 90-day moving mean FWI, FWI$_{95d}$–number of days of 95th percentile FWI, and FWI$_{max}$–annual maximum FWI) and region, with dots showing results for each of the nine climate models used. Black dots denote where the 95% confidence interval of the risk ratio exceeds 1. The box plot depicts the interquartile range with the black horizontal line showing the multi-model median. For reference, a risk ratio of 1 (dashed black line) indicates no change from the preindustrial period.

continuity stemming from fire suppression, abandonment of rural agriculture, forestry practices that favor high-density monocultures, and the introduction of flammable non-native species[44]—all of which favor fires that resist fire suppression coincident with extreme fire weather. Moreover, concurrent widespread fire weather extremes increase potential for favorable fire growth across broad regions that can feed back to limit the efficacy of fire suppression efforts[17].

Conversely, we found a different response for regions that were dominated by forest degradation (Amazon, Southeast Asia) and widespread anthropogenic fire use, including prescribed burning (e.g., southeastern US). Extreme fire years in tropical biomes qualitatively occurred in places subject to deforestation–agricultural burning that coincided with moderate-to-severe drought conditions. The tendency for most of the most extreme fire years seen in tropical ecoregions in the first decade of the study period (Fig. 1b) may be a product of waning deforestation rates[45]. Conversely, 2024—which was outside of our study period—had the highest burned area on the MODIS record for parts of the Amazon, coinciding with extreme drought and record chronic FWI[46]. Notably, we show here that Amazon was one of the global hotspots of escalating odds of FWI extremes with continued climate change.

The lack of unifying relationships between extreme fire years in forested areas and antecedent moisture in the prior one to two years is consistent with these regions being primarily flammability-limited and less dependent on factors influencing antecedent vegetation productivity[2,47]. Longer-term drought was not as closely linked to extreme fire years as within fire season FWI metrics, similar to findings in the western US[48]. Likewise, our focus on surface fire weather metrics does not account for the potential for other atmospheric drivers, such as instability that contribute to plume-dominated fires or widespread lightning events in ignition-limited fire regimes[49] facilitating extreme fire years. Lastly, while these results hold for forested ecoregions globally, there are distinctly different biophysical constraints in fuel-limited grassland and shrubland-dominated regions, suggesting that our findings between extreme FWI and extreme fire years are not transferable to non-forested landscapes.

We demonstrate that human-caused climate change has significantly increased the likelihood of extreme regional fire years in global forested areas. Prior work has shown that anthropogenic climate change has and is projected to increase fire weather for parts of the globe[36,50,51], and a few studies have focused on changing probabilities of extreme fire years, albeit at limited geographical scales[52]. Here we find that the likelihood of 1-in-15-year FWI extremes—comparable to the magnitudes coincident with the most extreme regional fire years this century—are 88-152% greater across forested ecoregions under contemporary climate (2011–2040) than under a quasi preindustrial climate (1851–1900) with robust results across the climate models used herein. Our findings largely align with limited regional attribution studies in terms of the direction of change[27,28,53], although they differ in magnitude due to varied choices of indicators, attribution approaches, spatial scales, models, and specific thresholds used (e.g., event-attribution studies often use thresholds based on events versus the generic thresholds used herein). Models with multiple ensemble members were used to aid in robustness. However, we are limited by only using output from nine climate models that had the necessary meteorological variables for calculating FWI and furthermore are unable to resolve finer-scale changes that might be pertinent to extreme fire weather or land-surface feedbacks, given the use of non-downscaled climate model output.

Our results are particularly important in the context of fire management—historically, response-driven fire management has focused on suppressing individual wildfires as they arise with the assumption that sufficient emergency response resources will be available to deal with both existing and new fires. Region-wide extreme fire years, however, have demonstrated the fallacy of this strategy, as suppression resources are overwhelmed by the high number of large wildfires burning simultaneously, and demand exceeds supply. While climate mitigation efforts to reduce greenhouse gas emissions remain crucial[2,12], adaptation strategies will be vital in improving readiness and resilience in at-risk regions. These on-the-ground efforts take several forms: proactive fuel reduction at local scales to reduce fire intensity and weaken climate-fire sensitivity, fire prevention efforts to minimize human-caused ignitions,

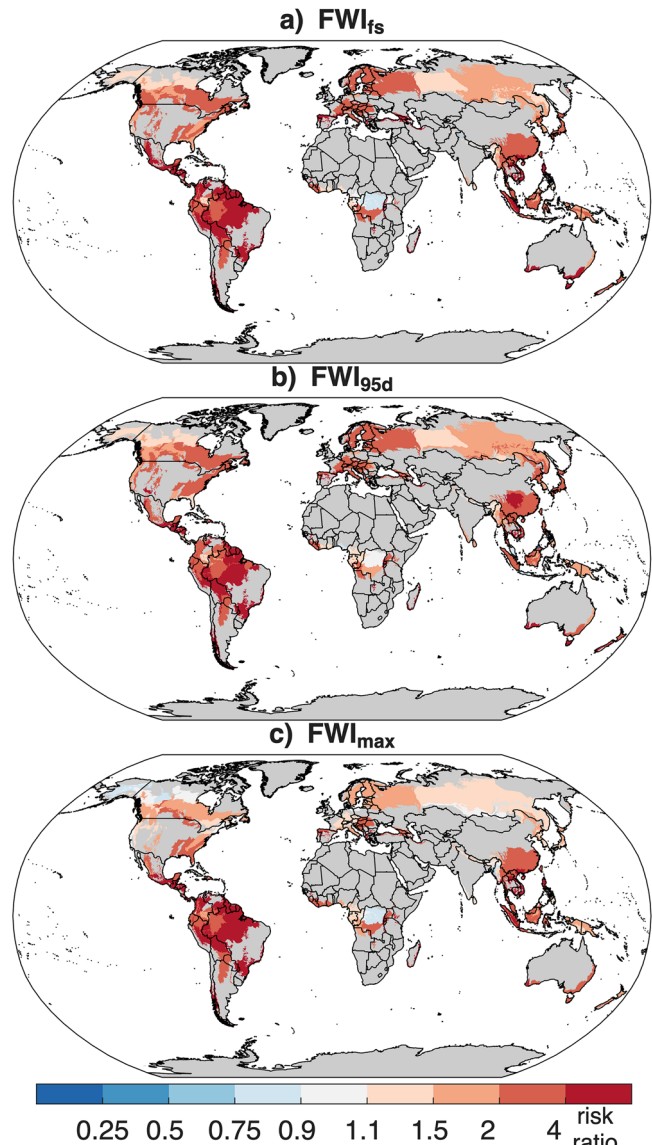

**Fig. 5 | Increased potential for climate-driven extreme fire years across forested ecoregions.** Nine-model median risk ratio for the Fire Weather Index (FWI) metrics based on historical 1-in-15 year event (per the 1979–2023 period) for (**a**) FWI$_{fs}$– maximum 90-day moving mean FWI, (**b**) FWI$_{95d}$–number of days of 95th percentile FWI, and (**c**) FWI$_{max}$–annual maximum FWI. This shows the ratio of current (2011–2040) versus quasi-preindustrial (1851–1900). For reference, a value of 1 indicates no change from the preindustrial control, while a value of 2 indicates a doubling in probability. Ecoregions with <20% forest land are excluded and shaded grey.

and adaptation measures to minimize direct and indirect impacts of fires on communities and ecosystems[54]. With increasingly extended fire seasons, resource limitations will necessitate a shift in fire management tactics to be less reactive and suppression-oriented and become more proactive in restoring beneficial fire and preparing landscapes and communities to withstand fire and mitigate disastrous outcomes.

## Methods
### Fire and climate datasets
Monthly burned area was sourced from MODIS MCD64A1[55] at 500 m horizontal resolution for March 2002–February 2024. To distinguish between tree and non-tree cover burned area, we employed the annual MODIS Vegetation Continuous Fields (MOD44B[56]) and defined forested burned areas where pre-fire tree cover exceeded 20%[16]. Annual forested burned area was calculated using fire years defined as March-February[57], which better captures the Southern Hemisphere fire season than calendar years and reflects the nadir in global fire activity in February-March. Note that the MODIS burned area product does not discern between wildland fire from other sources of biomass burning (e.g., agricultural burning, deforestation), and hence we intentionally do not refer to this as wildfire or wildfire burned area.

Additional fire datasets used included fire carbon emissions from the Global Fire Emissions Database (GFED4s[58]). GFED fire carbon emissions data were restricted to contributions from boreal and temperate forest fires and deforestation/degradation fires, and limited to the period 2002–2023; no data were available for the first 2 months of 2024. We additionally used attributes from individual fire events from the global fire atlas[59,60], based on MODIS MCD64A1, which was extended through February 2024. These data provide the number of fire events and fire sizes during March 2002–February 2024.

Meteorological data were acquired from ERA-5[61] at a 0.25° resolution from 1979–2024. To allow for direct compatibility with FWI calculated from climate model output, we calculated FWI using daily maximum temperature, minimum relative humidity, daily precipitation total, and daily mean wind speed as in prior studies[28,36] and used the overwintering procedure that adjusts for initial values as the start of the fire season. We considered both antecedent moisture in the prior one to two years, using actual evapotranspiration and precipitation, as well as concurrent fire danger realized through the FWI. We focus our analysis on the FWI from the Canadian Forest Fire Danger Rating System[62] given its widespread use and correlative relationships with regional interannual burned area globally[16] and extreme fire events[30]. FWI incorporates the influence of fuel drying across a stratum of three fuel classes along with potential spread rate, and is calculated as a function of temperature, relative humidity, precipitation accumulation, and wind speed.

Climate model output from nine models participating in CMIP6 was used in our climate attribution exercise. These data were acquired from Quilcaille et al., who calculated the FWI metrics used herein. We constrained our analysis to the historical (1851–2014) and SSP2-45 scenario (2015–2040) for models that calculated FWI using daily minimum relative humidity and had at least three ensemble members (Table S5). The latter criterion was used to improve the signal-to-noise ratio in calculations of risk ratio and is particularly useful for attribution of rare extremes. FWI output from climate models was not bias-corrected. While the direct use of climate model output inhibits direct comparison of numerical values of metrics like FWI with observations, this approach best preserves the climate change signal used herein[63].

### Identifying extreme regional fire years
There is a myriad of approaches for codifying extreme fires[31]. Arguably, the use of burned area in defining extreme fire years is among the more accessible and reproducible, given the availability of global burned area data from satellite records along with country-specific fire data for some geographies that dates back further. Herein, we define extreme regional fire years as those with the greatest amount of forested area burned over a given geographic area during the 2002–2023 observational period. However, we acknowledge that this working definition of extreme fire years is not a panacea for fire impacts, as burned area does not scale to societal fire impacts[33].

Our focus on extreme regional fire years presumes the dominance of top-down climate drivers and enablers of fire activity. We considered multiple scales of analysis. National boundaries are commonly used because fire records, impact reporting, and management strategies are typically organized at the country level. However, our study also examines larger scales—such as regions spanning multiple countries—and ecoregion-based scales that reflect shared macroscale vegetation regimes.

We first identified a small but representative sample of eight recent extreme fire years across the four major forest biomes of the world affected by fires: boreal, temperate conifer, temperate broadleaf, and tropical[32]. Within each forest biome, we selected the two extreme fire years (i.e., highest area burned in the MODIS period during 2002–2023) from different parts of the world that have been identified in recent literature. Through this process, we identified eight total case study events: (1) 2006 Southeast Asia, (2) 2010 Amazon Basin, (3) 2012 Siberia, (4) 2017 Mediterranean Basin/Iberian Peninsula, (5) 2019–2020 Southeast Australia, (6) 2020 western US, (7) 2023 Chile, and (8) 2023 Canada (Table 1; Fig. 1a). We complement these case studies by conducting analyses across forested ecoregions of the globe[64]. Following Abatzoglou et al., we excluded ecoregions where forested land accounted for <20% of the ecoregion.

For each region, we tabulated several statistics coincident with the peak fire year: (i) forested area burned from MCD64A1, (ii) fire C emissions from GFED4s, and (iii) both the total number of fires and very large fires from the global fire atlas[59,60]. For the case study regions, we use a threshold of 100 km$^2$ to delineate very large fires. However, as not all ecoregions experience fires exceeding 100 km$^2$, we define very large fires at the ecoregion level as the top 1% of fires by size for each ecoregion. We contextualize these statistics relative to the average annual values for all remaining years. For case study regions, we cataloged contributing climatic and non-climatic factors and fire impact data (e.g., fatalities, evacuation, structure loss, cost), where available, from the literature.

### Climatic context of regional extreme fire seasons

Following Abatzoglou et al., we calculated three annual FWI metrics that track different attributes of fire weather and their relationship to potential fire activity: (i) days per year exceeding the local historical (1979–2023) 95th percentile FWI (FWI$_{95d}$), (ii) annual maximum FWI (FWI$_{max}$), and (iii) annual maximum 90-day mean FWI (FWI$_{fs}$) given that a majority of regional annual burned area occurs in a continuous 2–5 month window during the year[16]. To complement our leading hypotheses that extreme fire years are enabled by extreme fire weather, we considered annual actual evapotranspiration and precipitation in the previous fire year (March–February) from ERA-5. In fuel-limited fire regimes, increased primary productivity during anomalously wet conditions in the prior growing season is linked with enhanced fire activity[2]. While antecedent moisture–burned area links are weak across forested ecoregions globally[16,47], historical large fire years in dry forests of the southwestern US often occurred 1–2 year after wet conditions[65].

To understand common climate factors associated with extreme regional fire years, we ranked the climate metrics, including concurrent FWI metrics and antecedent moisture, relative to the modern observational period of ERA-5 (1979–2023). A longer 45-year time series was examined here rather than strictly the contemporaneous 2002–2023 period to provide more robust estimates of extremes. We additionally estimated return period intervals coincident with extreme regional fire years for each metric. Return periods were estimated using a GEV approach using data from a 1979–2023 baseline. While studies often use non-stationarity extreme value analyses in a changing climate[66], typically using global mean temperature as a covariate, we opt not to add that layer of complexity in our core analysis. Instead, we use a GEV approach over specific time periods to assess the changing probabilities of extremes as done in some attribution studies[67,68], and to aid in the interpretability of our results.

### Quantify the role of human-caused climate change in contemporary extreme fire seasons

Next, we quantified how anthropogenic climate change has altered the likelihood of climate-enabled extreme fire years. This was done using a probabilistic approach common to climate attribution questions and guided by analytical results from our observational analyses. Specifically, we used the return intervals of FWI metrics informed by the GEV analysis of observed extreme fire years and applied this approach across all metrics and regions of analysis using common baseline years (1979–2023) as in observations.

We calculated the magnitude of the 15-year return period for FWI metrics by pooling data across all ensemble members per GCM for the baseline period (1979–2023) to match that of the observational results. We then calculated the likelihood of exceedance from this 15-year return period for FWI metrics for both a quasi-preindustrial period (1851–1900) and a contemporary period (here, 2011–2040). Note that we use a period centered around the present day rather than the recent past (e.g., 1991–2020), as our goal is to quantify the present-day likelihood of climate extremes that have been linked with recent extreme regional fire years. These estimates were then used to calculate risk ratios (RR) of FWI extremes for the contemporary period relative to the quasi-preindustrial period. These risk ratios provide a numerical measure of how the risk of extremes has changed due to anthropogenic climate change and are calculated as the ratio of the probability of FWI extremes exceeding the 1-in-15-year value (defined based on model years 1979–2023 to match observational analytics) under the contemporary climate to the quasi-preindustrial climate. Calculations were done separately for each GCM by pooling all model years across ensembles for the contemporary period (2011–2040) together and likewise all model years across ensembles for the counterfactual quasi preindustrial (1851–1900) period. We additionally used a bootstrap resampling ($n = 1000$) approach to quantify cases where the 95% confidence interval of RR exceeds 1. We present results for the 9-model median, except where otherwise specified. We note that our attribution relies on data-driven estimates of internal climate variability using early-industrial historical forcings with nominal anthropogenic contributions, rather than historical natural forcings—such as volcanic eruptions—that explicitly exclude human influence on the climate system.

## Data availability

Historical meteorological data used herein from ERA-5 are available from (https://cds.climate.copernicus.eu/), and climate projections used in the paper are available at https://www.research-collection.ethz.ch/handle/20.500.11850/583391. Gridded burned area data is available from the MODIS global burned area product MCD64A1 (https://modis-fire.umd.edu/ba.html). The Global Fire Atlas and GFED4s fire carbon emissions are available at https://zenodo.org/records/11400062 and https://www.geo.vu.nl/~gwerf/GFED/GFED4/, respectively.

## Code availability

No specialized code for data analysis was developed for this study.

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

## Acknowledgements

The work was supported through grants from the NSF Growing Con-vergence Research Program (OAI-2019762) to J.T.A. and A.C.C., Department of Interior's Joint Fire Science Program (21-2-01-1 and 21-2-01-3) to J.T.A., M.S., and E.W. and UK Natural Environment Research Council (NERC) grant NE/ V01417X/1 to M.W.J.

## Author contributions

J.T.A., M.W.J., and C.A.K. conceived the study. J.T.A. wrote the first draft of the paper. J.T.A. conducted all analyses. J.T.A., C.A.K., A.C.C., M.S., E.W., M.T., and M.W.J. contributed to the study design, results assess-ment and interpretation, and writing of the paper.

## Competing interests

The authors declare no competing interests.
