## [Transparent Peer Review file · Nature Communications]

Climate change has increased the odds of extreme regional forest fire years globally

Corresponding Author: Dr John Abatzoglou

Version 0:

Reviewer comments:

Reviewer #1

(Remarks to the Author)

This manuscript considers global ecoregions and particular regional case studies to characterize the fire weather during extreme fire years, followed by the change in likelihood of the 1-in-15 year extreme fire weather between the current and pre-industrial climates.

This manuscript is well written and the figures are mostly easy to understand. I have some methodology concerns which I have described below; in particular, I think more care could be given to the attribution approach.

Major comments

1. GEV fits. First, how is a GEV distribution fit to the FWI_95d metric, which is not block maxima? Second, lines 334-336 seem to imply that the stationary GEV is most often used in climate attribution and I think this gives the wrong impression. Many event attribution studies do use non-stationary GEV distributions (with global mean temperature as the covariate). Alternatively, event attribution studies that use stationary GEV distributions do so for large ensembles of model simulations (so still enough data to get a robust fit) over a short time window (assume stationarity over 10 or 20 years, for example). In the current analysis, the climate from 1979-2023 is assumed to be stationary; I would suggest checking trends in the relevant variables to verify the validity of this approach.

2. Risk Ratio calculation. Uncertainty on the RR seems to be neglected in this analysis. It is very common across event attribution studies to use a bootstrap resampling approach to estimate an uncertainty range on the RR, though other methods are available. Considering the small RR presented here, I wonder if the uncertainty range on these estimates is fully greater than 1.

3. A model evaluation step seems to be missing. And although the authors acknowledge different climate sensitivities of the climate models used in the analysis, the impact of this on the results was not well discussed. To reduce uncertainty from the different model warming rates, two papers already cited in this manuscript offer two slightly different but complementary approaches: van Oldenborgh et al. for the Australia event used a non-stationary GEV as a function of global mean temperature, so the observed temperature increase for the current climate can be used to extract the appropriate probabilities. In Kirchmeier-Young et al. for the Canada event, a global warming level approach was used, selecting the decade from each model that corresponds to the observed level of warming before calculating probabilities from the pool of realizations with a stationary approach. I encourage the authors to consider a similar approach, or at the least, discuss the added uncertainty in the analysis.

Minor comments

1. Suggest using "RR" instead of rho to denote the risk ratio as this notation is widely used across the literature.
2. Multiple figure captions refer to the "change in risk ratio" but I think this should be just "risk ratio." The risk ratio itself is the change in likelihood (comparing the current to pre-industrial climates) rather than a risk ratio calculated in each climate and then compared (as the current phrasing would imply).
3. FWI calculation. More details on how the FWI was calculated from the reanalysis and climate models are needed. As described, it is difficult to tell if the two datasets were processed in the same way. Is hourly data used from ERA5 (and local noon values input into the FWI calculation) or daily data to match the models? Furthermore, did the calculation consider overwintering, which can be important for some regions?

4. There are a few additional studies looking at attribution of fire weather globally the authors may consider including: Liu et al. 2022 in Climatic Change (doi: 10.1007/s10584-022-03409-9); Touma et al. 2021 in Nature Comms (doi: 10.1038/s41467-020-20570-w)
5. Line 138: For the Canada region in particular, which is made up many ecozones, the signal seems to show strong regional variation (as shown in Figure 5). Perhaps a mean over the larger region is less useful here.
6. Line 142: Suggest clarifying what this 85-150% means in terms of the risk ratios presented (presumably 1.85 - 2.5). Also here and elsewhere this range is reported (including in the abstract), suggest clarifying that this range is across metrics and not implying an uncertainty range.
7. Line 157: Suggest considering noting the influence of anthropogenic climate change on wind extremes is more uncertain instead
8. Line 219-221: This is very important and I'm glad to see it mentioned here. It might also be worth adding that the analysis in this paper uses a generic threshold rather than one defined based on the observed event.
9. Line 284: Is it correct to interpret this as each region only has one extreme fire year which is the year with the largest area burned in the analysis period?
10. Line 292: Is there a reference for the ecoregions?
11. Line 346-349: This description is difficult to follow. The return period for which metrics is calculated from 1979-2023? Is it that the magnitude of the 15-year event is being determined from that base period and then the probability of that event is calculated for the current and pre-industrial climates? If so, suggest saying this explicitly.
12. Figure 1: The Siberia region highlighted in panel a shows 2022/2023 for the highest year in panel b, but 2012 is noted in panel c. Suggest adding a brief explanation to reconcile this.
13. Figure 1: The scale in panel b is confusing as the colors presumably apply to two years but are labeled for one year. Perhaps labelling at the boundaries of colors would help.
14. Figure 1: Is the ET metric applied on the opposite scale, such that being at the bottom of the years is more extreme than being at the top? Even understanding that drier means higher fire risk, there might be less confusion if the metric were flipped here such that Chile's blue dot would actually be red.
15. Table 1: Could the (forested) area of each of these regions be included? Also suggest noting in the text the challenge with comparing RRs between regions of such different scales.
16. Figure S7: The caption notes a median RR across the ensemble, but the methods section implies all ensemble members were pooled for the calculation. Clarity on the method is needed and pooling would be more robust than estimating probabilities within each realization individually.
17. Table S1: It seems most of the information in this table is already in Table 1. Suggest combining.

Reviewer #2

(Remarks to the Author)

I am happy to report that this manuscript is very well-written and includes novel findings of importance to the global forest ecology and carbon cycle science community. I am especially impressed with the explanation of the nuance associated with increasing fire activity and how the authors have carefully been able to deal with attribution. This is important because being able to separate management (e.g., fire suppression, agricultural burning) from anthropogenic climate change as causes for increased fire activity is essential for developing mitigation strategies. My one suggestion for this manuscript would be to highlight a few more of the salient (and rather impactful) results in the abstract. For example, "...more substantial increases in very large fires (319% median increase)...Likewise, carbon emissions from forest and deforestation sources were many times higher (416% median increase)..." could be highlighted in the abstract.

Reviewer #3

(Remarks to the Author)

This manuscript aims to assess the impact of climate change on the frequency of regional fire incidents globally.

GENERAL COMMENTS

The manuscript is well written, but it has several problems.

The most important is the lack of novelty. The scientific community that studies wildfires knows that extreme wildfire incidence (number of wildfires and burned area) only occurs in anomalous atmospheric and climatic conditions that are also very extreme and that a large number of large wildfires/burned area (occurring in extreme conditions) necessarily leads to major consequences of wildfires, including gas emissions. Therefore, the question that the authors have to clarify is what is new in this article that the reader could not obtain from the various articles published on the episodes/cases of this study?

It's certainly not the databases or the methodologies. It is also not the approach, as studying this problem on an annual scale is very similar to studying one of the large fires or the episode of large fires that tend to occur in these extreme atmospheric/climatic conditions, especially with the metrics chosen by the authors. These metrics that the authors select to study years of extreme regional fire are very dependent on the conditions during high incidence episodes, as they are the annual maximum daily FWI, number of days with FWI above P95 (calculated for the entire period, and the FWI of the "90-day fire season".

Another problem with the manuscript is the quality of the figures. Almost all of them need a lot of additional work to be at an acceptable level and self-explanatory. Please see comments and suggestions below. The manuscript is unclear and

rigorous in some fundamental aspects. For example, is this article about fire (desired, controlled, and desired combustion) or about wildfires (unwanted, uncontrolled, and unauthorized combustion)? I know that it is traditional to use the concept "fire" to describe characteristics of the phenomenon under study (such as fire regime, fire incidence, fire severity, etc.) and certainly the database includes events in which fire is used as a tool, but it seems clear to me that the manuscript is essentially about wildfires and that it is necessary to evolve to more correct and appropriate concepts; at least these issues must be discussed;

Some of the variables used and studied are not defined, at least adequately defined. For example, how was the number of fires shown in various figures calculated?

How is the risk ratio calculated? A clear and objective definition is necessary, especially because the authors indiscriminately refer to risk ratio and odds ratio, while risk ratio, odds ratio and hazard ratio are different concepts.

The manuscript also does not adequately or completely discuss what the problems and limitations of their approaches are and, consequently, the robustness of their results and conclusions. In addition to the problems affecting the quality of the data, several issues should be addressed by the authors. Some of the (usual) questions are, for example, why did you choose 9 simulations? Why those 9 models (Table S3)? What is the danger for the potential significance of the results of using fire data for a relatively short data period?

SPECIFIC COMMENTS

Line 1: Risk and Odds ratio are not the same. However, the authors treat them as the same thing.

Lines 69-70. This sentence is not true. There are papers defining extreme wildfire events. Please see, for example, Tedim, F., Leone, V., Amraoui, M., Bouillon, C., Coughlan, M. R., Delogu, G. M., ... & Xanthopoulos, G. (2018). Defining extreme wildfire events: Difficulties, challenges, and impacts. *Fire*, 1(1), 9.

Line 244: Please, include a space or non-breaking space between 500 and "m". Please check all other units.

Line 352: The risk ratio is not defined. A clear definition of risk ratio is needed.

Tables & Figures

The data and results presentation must be more rigorous. The Tables and Figures must be self-explanatory, and the Figures of this manuscript are not.

Table 1

The reader may interpret MODIS as the burned area, GFA as the number of fires and also the number of "very large" fires, and GFED4s as the carbon emissions. The authors should clarify the relationship between the variables and the acronyms, otherwise, it seems to be an unnecessary and confusing definition of acronyms.

The "number of fires $\geq 100 \text{ km}^2$ " should be "number of fires with burned area greater or equal to 100 km^2 " or something similar.

Unnecessary and confusing definition of acronyms. Why define acronyms (e.g., GFA) not used in the Table? What is the relationship between "the number of fires" and "GFA"? Why define/mention GFA for both the number of fires and the number of fires with $\text{BA} \geq 100 \text{ km}^2$?

Anomalies are defined differently. In the title, "Anomalies" are "% of average for non-extreme years", while in the caption, "Anomalies are compared concerning the mean calculated from all remaining years ...". These definitions may or may not be the same. Authors must clarify what an extreme year is. How many extreme years are defined?

The use of * and + is confusing. These characters should be in superscript, otherwise $100+$ can be more than 100.

Table S1

Why is this table needed? Why don't you merge the information from Table 1 and Table S1?

Why use a different citation style in the Table line for Chile?

Figures

Figure 1

Do not use the minus sign to define period; use n-dash. Please check and correct the entire manuscript.

A title in each panel will increase the readability of the Figure

Panel c, why circles of different sizes? What is the meaning of different sizes?

It seems like there is an unfinished sentence "Quantitative ranking of three FWI metrics (FWI_f: max 90d moving mean FWI during the year; FWL95: # of days of 95th percentile FWI, FWI_{max}: maximum daily FWI in the year) coincident to"

All acronyms must be defined, including FWI.

Figure 2

The title of this Figure does not seem suitable. The Figure only presents estimates of the return periods of extreme FWI metrics.

The titles of each panel do not seem suitable either. It seems that you are presenting FWI_{fs}, FWI_{95d} and FWI_{max}.

All acronyms must be defined.

Figure 1 and Figure describe differently the areas in grey. Please harmonize.

Figure 3

The titles of each panel do not seem suitable either. It seems that you are presenting the fire characteristics and not the anomalies of FWI_{fs}, FWI_{95d} and FWI_{max}.

Figure 4

How are the Box plots defined? Did you use P90, P75, P50, Mean, P25, P75, P10, variance, IQR?

Figure 5

As in Figure 3, the titles of each panel do not seem suitable either. It seems that you are presenting the fire characteristics and not the median risk ratio of FWI_fs, FWI_95d and FWI_max.

Please check and correct the figures in the supplementary material based on previous comments and suggestions.

Version 1:

Reviewer comments:

Reviewer #1

(Remarks to the Author)

I appreciate the authors' consideration of my comments and I am generally satisfied with how they have been addressed. I do not have any additional comments that would delay publication.

Reviewer #3

(Remarks to the Author)

Dear authors,

I appreciate your response to my comments, questions, and suggestions. I am convinced that you have appropriately edited the document to correct errors and typos, clarified what needed clarification, and answered my deeper questions, correctly defending your position and decisions.

I believe that the manuscript is acceptable for publication.

Best regards, Mário

Response to Reviewer #1

Reviewer 1 had several helpful suggestions for clarifying the approaches, including some suggested language which we adopted in the revision. There were several useful technical questions from the reviewer that we have added to the manuscript to improve upon the initial submission and aid in interpreting the robustness of the results. We appreciate their input and have made changes to the manuscript to address their suggestions.

Reviewer #1 (Remarks to the Author):

This manuscript considers global ecoregions and particular regional case studies to characterize the fire weather during extreme fire years, followed by the change in likelihood of the 1-in-15 year extreme fire weather between the current and pre-industrial climates.

This manuscript is well written and the figures are mostly easy to understand. I have some methodology concerns which I have described below; in particular, I think more care could be given to the attribution approach.

Major comments

1. GEV fits. First, how is a GEV distribution fit to the FWI_95d metric, which is not block maxima? Second, lines 334-336 seem to imply that the stationary GEV is most often used in climate attribution and I think this gives the wrong impression. Many event attribution studies do use non-stationary GEV distributions (with global mean temperature as the covariate). Alternatively, event attribution studies that use stationary GEV distributions do so for large ensembles of model simulations (so still enough data to get a robust fit) over a short time window (assume stationarity over 10 or 20 years, for example). In the current analysis, the climate from 1979-2023 is assumed to be stationary; I would suggest checking trends in the relevant variables to verify the validity of this approach.

We appreciate this insightful comment. First, we apply a GEV distribution fit using the maximum likelihood method to model the tails of various distributions, not exclusively those from block maxima. While FWImax and FWI90d are traditional block maxima types, we also run the GEV for FWI_95d by summing the number of days per year exceeding the 95th percentile into an annual sum, making it a block maxima-type metric. GEV is typically reserved for block maxima, but other studies have occasionally performed the approach with metrics such as annual means (e.g., Vogel 1996), among others. Counts of the number of days per year exceeding the 95th percentile are well resolved by GEV (per distribution fit tests), and hence we feel confident that this approach is generalizable to capture such extremes.

Vogel, R.M. and Wilson, I., 1996. Probability distribution of annual maximum, mean, and minimum streamflows in the United States. *Journal of hydrologic Engineering*, 1(2), pp.69-76

Second, we acknowledge that our wording was not accurate in the original submission, as many studies have used non-stationary GEV distributions using GMST as a covariate. We have modified this portion of the text to avoid confusion. We opted not to run non-stationary GEV given the types of problem we are trying to address – namely, in quantifying the unusual nature of historical FWI extremes during extreme fire seasons, and the somewhat simpler interpretation of results without needing to invoke nonstationary extreme value analysis. Related to this point and point 3 below, we do not find strong evidence to suggest significant sensitivity of our results to changes in GMST, at least across variability in climate models for the time frame considered in this work.

“While studies often use non-stationarity extreme value analyses in a changing climate⁶⁶ typically using global mean temperature as a covariate, we opt not to add that layer of complexity in our core analysis. Instead, we use a GEV approach over specific periods to assess the changing probabilities of extremes as done in some attribution studies^{67,68}, and to aid in the interpretability of our results.”

2. Risk Ratio calculation. Uncertainty on the RR seems to be neglected in this analysis. It is very common across event attribution studies to use a bootstrap resampling approach to estimate an uncertainty range on the RR, though other methods are available. Considering the small RR presented here, I wonder if the uncertainty range on these estimates is fully greater than 1.

Indeed, we did not fully capture uncertainty in RR. Such efforts are commonly performed for event-attribution, but harder to operationalize and visualize for the broad set of geographies herein. Likewise, there are numerous ways to quantify uncertainty, including that from climate models (which we do provide in the manuscript), which further confounds issues. We have now added this information for the case studies events (Fig. 4; Figs. S7). Specifically, we use a bootstrap resampling of the contemporary climate ($n=1000$) and quantify results as being significant where the 95% confidence interval of the risk ratio did not include 1. For the most part, at least for our larger case study areas, RR values generally exceeding 1.5 were deemed significant. This is likely a byproduct of the pooling of multiple ensembles, as we expect that uncertainties would be higher for single-member estimates. It may also be a product of the longer time scales used in our analysis (e.g., using 30-year blocks). We opted not to undertake this uncertainty analysis for the global ecoregions, given the challenges in communicating such information in a meaningful way. However, text was incorporated to address this issue:

“Calculations were done separately for each GCM by pooling all model years across ensembles for contemporary period (2011-2040) together and likewise all model years across ensembles for the counterfactual quasi pre-industrial (1851-1900) period. We additionally use a bootstrap resampling ($n=1000$) approach to quantify cases where the 95% confidence interval of RR exceeds 1.”

3. A model evaluation step seems to be missing. And although the authors acknowledge different climate sensitivities of the climate models used in the analysis, the impact of this on the results was not well discussed. To reduce uncertainty from the different model warming rates, two papers already cited in this manuscript offer two slightly different but complementary approaches: van Oldenborgh et al. for the Australia event used a non-stationary GEV as a function of global mean temperature, so the

observed temperature increase for the current climate can be used to extract the appropriate probabilities. In Kirchmeier-Young et al. for the Canada event, a global warming level approach was used, selecting the decade from each model that corresponds to the observed level of warming before calculating probabilities from the pool of realizations with a stationary approach. I encourage the authors to consider a similar approach, or at the least, discuss the added uncertainty in the analysis.

We appreciate this comment, and note that a model evaluation step here (e.g., how credible are models are capturing attributes of FWI) may not be entirely necessary and may be somewhat challenging given that we are doing this globally meaning that some models may perform well in some areas and not others in terms of credibly capturing attributes of FWI and their trends. Understandably, for a regional analysis or event attribution, such efforts seem useful to incorporate. However, doing this for global FWI is beyond the scope of our analysis. We contend that we do already attempt to account for uncertainty by using several climate models, as our results focus on multi-model median results (Figs 4-5). Likewise, we do report variability across models (in Fig 4 and Fig S7). We have incorporated the ideas from the Kirchmeier-Young et al. study in our modified version of Figure S7. We now order the models in terms of the change in GMT between the quasi pre-industrial (1851-1900) and contemporary (2011-2040) periods. At least for the case studies, there are no systematic differences in RR as a function of changes in GMT. In addition, we now report global summaries of median RR and the percent of forested lands where $RR > 2$ and $RR < 1$ in Tables S3-S5. We briefly report on this in the text.

“Lastly, no systematic differences in RR were seen as a function of climate sensitivity or the increase in global mean temperature through 2040 (Fig. S7).”

Interestingly, we find weak evidence of somewhat higher RR for models that show lesser change in GMT (Fig R1 below). However, given the limited sample sizes, we do not take this analysis any further. These results, in combination with those in Fig S7, don't provide any compelling evidence supporting the idea that models with greater rates of warming show higher changes in the metrics we examined here. We also looked at relationships by equilibrium climate sensitivity and found similar negative results.

Figure R1: Scatterplot of changes in median RR across forested lands versus changes in global mean temperature between quasi pre-industrial (1851-1900) and contemporary (2011-2040) periods for (a) FWI_{fs}, (b) FWI_{95d}, and (c) FWI_{max}.

Minor comments

1. Suggest using “RR” instead of rho to denote the risk ratio as this notation is widely used across the literature.

Thank you for this suggestion. We have made this change to conform with the notation widely used in the literature.

2. Multiple figure captions refer to the “change in risk ratio” but I think this should be just “risk ratio.” The risk ratio itself is the change in likelihood (comparing the current to pre-industrial climates) rather than a risk ratio calculated in each climate and then compared (as the current phrasing would imply).

Good point, and thank you for catching this. We have made changes to the figure captions in a couple of places and a spot in the text.

3. FWI calculation. More details on how the FWI was calculated from the reanalysis and climate models are needed. As described, it is difficult to tell if the two datasets were processed in the same way. Is hourly data used from ERA5 (and local noon values input into the FWI calculation) or daily data to match the models? Furthermore, did the calculation consider overwintering, which can be important for some regions?

Thank you for raising this issue. We had been using the off-the-shelf FWI provided by Copernicus. However, their calculations do not include overwintering in CFFDRS calculations, which does indeed create incompatibilities with the CMIP6 FWI calculations. To better harmonize comparisons across data, we now calculate FWI using the same daily summary layers – specifically using maximum temperature, minimum relative humidity, daily precipitation total, and mean wind speed in ERA5 – as in the CMIP6

projections and overwinter DC in CFFDRS. These changes had very nominal differences in our results, but internal consistency in the use of observed and climate projections was a thoughtful suggestion.

4. There are a few additional studies looking at attribution of fire weather globally the authors may consider including: Liu et al. 2022 in Climatic Change (doi: 10.1007/s10584-022-03409-9); Touma et al. 2021 in Nature Comms (doi: 10.1038/s41467-020-20570-w)

Thank you, we had added these references in the discussion for context.

5. Line 138: For the Canada region in particular, which is made up many ecozones, the signal seems to show strong regional variation (as shown in Figure 5). Perhaps a mean over the larger region is less useful here.

Yes, this is certainly true and perhaps an issue for other zonation choices. Likewise, the spatial scales and spatial autocorrelation of climate realized through FWI may also impact fire-climate relationships that can obfuscate relationships. Our case studies were formulated by the literature, which has a tendency to lump things by zones that can span large regions (e.g., Canada). We feel that by providing the analyses also at the ecoregion level, we largely address this concern. However, to this point, we acknowledge that the ecoregions have a wide range of spatial extents that somewhat impact signal-to-noise ratios with RR. We added a point to this in the results:

“The same factors somewhat limit an fair comparison of RR across ecoregions due to the variety in their spatial scales.”

6. Line 142: Suggest clarifying what this 85-150% means in terms of the risk ratios presented (presumably 1.85 - 2.5). Also here and elsewhere this range is reported (including in the abstract), suggest clarifying that this range is across metrics and not implying an uncertainty range.

Thank you for catching this. Indeed, you are correct that these were not written accurately and instead refer to risk ratios as stated. We have revised this as follows:

“More generally across global forest ecoregions, the median RR of extreme FWI metrics was 1.88-2.52 across metrics, suggesting much higher likelihoods of such extremes under a contemporary climate than under a quasi pre-industrial climate (Fig. 5; Tables S1; S3-5 for individual models).”

7. Line 157: Suggest considering noting the influence of anthropogenic climate change on wind extremes is more uncertain instead

Fair point. This has now been modified to read:

“Many of the most extreme fire weather days realized through FWI_{max} are associated with wind extremes concurrent with dry fuels, the former of which exhibits more uncertainty to anthropogenic climate forcing³⁶.”

8. Line 219-221: This is very important and I’m glad to see it mentioned here. It might also be worth adding that the analysis in this paper uses a generic threshold rather than one defined based on the observed event.

As these studies have been event-based, it is completely reasonable to expect numerical values to differ from the generic approach applied to different regions here versus ones identified in each study. We modified this to read:

“Our findings largely align with regional attribution studies in terms of the direction of change^{27,28,52}, although differ in magnitude due to varied choices of indicators, attribution approaches, spatial scales, models, and specific thresholds used (e.g., event-attribution studies often use thresholds based on events versus the generic thresholds used herein).”

9. Line 284: Is it correct to interpret this as each region only has one extreme fire year which is the year with the largest area burned in the analysis period?

This is correct, we define the most extreme fire year as the year with the highest forest BA in the MODIS record. We have this stated in the fourth paragraph of the paper, where we lay out the working definition of extreme fire year:

“While a singular definition of extreme in fire science is absent^{31,33}, herein we adopt the term “extreme fire years” as those with the greatest amount of burned area in the MODIS record (2002-2023).”

10. Line 292: Is there a reference for the ecoregions?

In the methods section, about 10 lines further down, we highlight the Olsen ecoregions and provide a reference.

11. Line 346-349: This description is difficult to follow. The return period for which metrics is calculated from 1979-2023? Is it that the magnitude of the 15-year event is being determined from that base period and then the probability of that event is calculated for the current and pre-industrial climates? If so, suggest saying this explicitly.

This has now been revised to hopefully aid in readability:

“We calculated the magnitude of the 15-year return period for FWI metrics by pooling data across all ensemble members per GCM for the baseline period (1979–2023) to match that of the observational

results. We then calculated the likelihood of exceedance from this 15-year return period for FWI metrics for both a quasi-preindustrial period (1851–1900) and a contemporary period (here, 2011–2040)."

12. Figure 1: The Siberia region highlighted in panel a shows 2022/2023 for the highest year in panel b, but 2012 is noted in panel c. Suggest adding a brief explanation to reconcile this.

Technically, 2012 had a greater burned area than 2022 for the region identified. It is true that there are ecoregions within the Siberia gridbox that had their largest ecoregion burned area in 2022, but the region as a whole had slightly greater burned area in 2012. One can refer to the Figs S1-S5 to see just how marginally higher BA was in 2012.

13. Figure 1: The scale in panel b is confusing as the colors presumably apply to two years but are labeled for one year. Perhaps labelling at the boundaries of colors would help.

Yes, this is a good point. We have modified the scale accordingly so that both years are reflected in the scale.

14. Figure 1: Is the ET metric applied on the opposite scale, such that being at the bottom of the years is more extreme than being at the top? Even understanding that drier means higher fire risk, there might be less confusion if the metric were flipped here such that Chile's blue dot would actually be red.

We understand the reviewers' concern here. We have expanded this visual to now include ET and precipitation from 2-years prior and 1-year prior. In cases of chronic multi-year drought, very low precipitation and ET in the preceding years (here represented by cool colors) would appear. However, studies have also suggested that wet antecedent conditions one-to-two years prior increase fine fuel loading and large fire potential – primarily in fuel-limited regions. As such factors could conceivably influence burned area in forested regions – particularly in dry forests, we included this in our analysis. Given the duality of moisture in enabling fire, and the notation in our legend, we have decided to keep the colors as is.

15. Table 1: Could the (forested) area of each of these regions be included? Also suggest noting in the text the challenge with comparing RRs between regions of such different scales.

Agreed that there is value in caveating comparisons of RR or climate-extreme fire relationships across different scales. We added to this point in the results section

"The more muted change in RR shown here compared to case study regions is likely a product of the smaller spatial scales of ecoregions that weaken signal-to-noise ratios. The same factors somewhat limit a fair comparison of RR across ecoregions due to the variety in their spatial scales."

We opt not to include the extent of forest for these regions in the interest of space, as we feel the table is already quite full.

16. Figure S7: The caption notes a median RR across the ensemble, but the methods section implies all ensemble members were pooled for the calculation. Clarity on the method is needed and pooling would be more robust than estimating probabilities within each realization individually.

Thank you for noting this. We had tried both approaches initially. We added the following sentence to the methods to clarify:

“We calculated the magnitude of the 15-year return period for FWI metrics by pooling data across all ensemble members per GCM for the baseline period (1979–2023) to match that of the observational results.”

17. Table S1: It seems most of the information in this table is already in Table 1. Suggest combining.

We agree. This information was decoupled from Table 1 since it became very difficult to integrate into a single Table with all references. This has now been addressed and contained within Table 1.

Response to Reviewer #2

We thank Reviewer 2 for their positive review and how the results may add value to the global forest ecology and carbon cycle scientific communities. We believe that our key results, that extreme fire years in forested systems nearly universally coincide with unusual-to-extreme fire weather conditions, do help highlight the dominant top-down weather-climate drivers. The bottom-up influences of fire suppression, fuel management, ignitions, and agricultural practices are certainly important for fire regimes, but perhaps less so for the phenomena we focus on here.

Reviewer #2: I am happy to report that this manuscript is very well-written and includes novel findings of importance to the global forest ecology and carbon cycle science community. I am especially impressed with the explanation of the nuance associated with increasing fire activity and how the authors have carefully been able to deal with attribution. This is important because being able to separate management (e.g., fire suppression, agricultural burning) from anthropogenic climate change as causes for increased fire activity is essential for developing mitigation strategies. My one suggestion for this manuscript would be to highlight a few more of the salient (and rather impactful) results in the abstract. For example, "...more substantial increases in very large fires (319% median increase)...Likewise, carbon emissions from forest and deforestation sources were many times higher (416% median increase)..." could be highlighted in the abstract.

The abstracts for this journal are very short (150 words), which limits the ability to fairly spell everything out. We do have the following in the abstract that should address this suggestion:

"These extreme years commonly coincided with extreme (1-in-15-year) fire weather indices (FWI) and featured a four and five-fold increase in the number of large fires and fire carbon emissions, respectively, compared with non-extreme years."

Response to Reviewer #3

We thank Reviewer 3 for their thorough review of the manuscript. They highlighted several items for improving the manuscript and figures that we appreciate greatly. We have additionally added several clarifications to their points and better emphasized the novelty of the study in our revision. We thank the reviewer for their perspectives, insights, and care taking to raise the bar on this effort.

This manuscript aims to assess the impact of climate change on the frequency of regional fire incidents globally.

GENERAL COMMENTS

The manuscript is well written, but it has several problems.

The most important is the lack of novelty. The scientific community that studies wildfires knows that extreme wildfire incidence (number of wildfires and burned area) only occurs in anomalous atmospheric and climatic conditions that are also very extreme and that a large number of large wildfires/burned area (occurring in extreme conditions) necessarily leads to major consequences of wildfires, including gas emissions. Therefore, the question that the authors have to clarify is what is new in this article that the reader could not obtain from the various articles published on the episodes/cases of this study?

It's certainly not the databases or the methodologies. It is also not the approach, as studying this problem on an annual scale is very similar to studying one of the large fires or the episode of large fires that tend to occur in these extreme atmospheric/climatic conditions, especially with the metrics chosen by the authors. These metrics that the authors select to study years of extreme regional fire are very dependent on the conditions during high incidence episodes, as they are the annual maximum daily FWI, number of days with FWI above P95 (calculated for the entire period, and the FWI of the "90-day fire season".

We respectfully disagree with the reviewer on the issue of novelty - no study to date has attempted to examine contributing factors to extreme fire years/seasons at the global scale, nor evaluated how such extremes may be influenced to date by ongoing human-caused climate change. Indeed, there are several case studies that have performed event attribution for high-profile extreme fire seasons and fires (many of which are cited in the introduction), but the scales over which these case studies were applied have been limited, and also the scalability of the approaches used has not been evaluated. Certainly, this is the first effort to attribute the most extreme fire years/seasons of every forest region globally. As such, we contend that this paper addresses a key gap in the literature between event attribution focused on one area and general results that may be applicable across different parts of the globe, including in areas that are poorly studied.

We do agree with the reviewer that our results tend to agree with prior case studies in that extreme fire events and years co-occur with exceptional atmospheric conditions such as extreme fire danger. This

was the working hypothesis – but should not be conflated with a lack of novelty as our results extend beyond earlier work. Along these lines, we would note that fire is a coupled human-natural system such that human pressures may also facilitate or drive years with extreme fire activity in some fire regimes. However, no known studies have codified this globally for either contemporary extremes or how such extremes have changed with human-caused climate change. We contend that our results add value by demonstrating the nearly ubiquitous role of extreme fire weather (all else held equal), enabling and driving extreme fire seasons across the globe, as well as quantifying changes in such extremes using climate models.

We have added clarifying statements on novelty into the introduction and discussion to clarify these points.

Another problem with the manuscript is the quality of the figures. Almost all of them need a lot of additional work to be at an acceptable level and self-explanatory. Please see comments and suggestions below.

Thank you for the suggestions on figures. These are somewhat complicated visuals, and while reviewer 1 supported our use of visuals, we have tried to change them per the suggestions where we agreed that changes could improve the interpretability of results. We have also added more details in the figure captions so that they can be stand-alone, per the reviewers' suggestion.

The manuscript is unclear and rigorous in some fundamental aspects. For example, is this article about fire (desired, controlled, and desired combustion) or about wildfires (unwanted, uncontrolled, and unauthorized combustion)? I know that it is traditional to use the concept “fire” to describe characteristics of the phenomenon under study (such as fire regime, fire incidence, fire severity, etc.) and certainly the database includes events in which fire is used as a tool, but it seems clear to me that the manuscript is essentially about wildfires and that it is necessary to evolve to more correct and appropriate concepts; at least these issues must be discussed;

We intentionally include all burned area (herein referred to as fire) based on the MODIS data, as there is no commonly accepted practice to discern wildfire from controlled fires across global lands. Indeed, one would expect wildfire to exhibit a stronger response to climate, and such relationships have been alluded to in previous studies, whereby weak climate-fire relationships may occur in landscapes with abundant controlled or prescribed fire (e.g., Abatzoglou et al., 2018). We refrain from calling this wildfire since not all burned area is wildfire, and hence we intentionally keep the term fire and burned area throughout. Our working hypothesis is that extreme fire seasons are rarely only a function of excess intentional fire. In some regions, agricultural/deforestation fires can escape with dry conditions and become uncontrolled fires that can lead to significant increases in burned area. Nonetheless, we agree that this point should be better stated in the manuscript for readers less familiar with the datasets and caveats. We have added this as follows:

“Note that the MODIS burned area product does not discern between wildland fire from other sources of biomass burning (e.g., agricultural burning, deforestation).”

Some of the variables used and studied are not defined, at least adequately defined. For example, how was the number of fires shown in various figures calculated?

We have gone through and attempted to better define the variables in the methods section. For the specific question here, the number of fires was calculated using the Global Fire Atlas data, which tracks individual fires from MODIS. Similar exercises are performed for the number of large fires using fire size thresholds ($\geq 100\text{km}^2$ for the case study regions, and top 1% of fire defined by ecoregion for the ecoregions).

How is the risk ratio calculated? A clear and objective definition is necessary, especially because the authors indiscriminately refer to risk ratio and odds ratio, while risk ratio, odds ratio and hazard ratio are different concepts.

Thank you for pointing this out. We did not adequately define the risk ratio in the manuscript. We have better defined the risk ratio in the revision to hopefully clarify this. We have now added a brief definition of risk ratio in the main text where it is first referenced.

“The RR is calculated as the ratio of the probability of FWI extremes exceeding the benchmark 1-in-15-year value under the contemporary climate to the counterfactual (here quasi-preindustrial) climate.”

Note that we only use risk ratio in the paper and do not refer to odds ratio or hazard ratio.

The manuscript also does not adequately or completely discuss what the problems and limitations of their approaches are and, consequently, the robustness of their results and conclusions. In addition to the problems affecting the quality of the data, several issues should be addressed by the authors. Some of the (usual) questions are, for example, why did you choose 9 simulations? Why those 9 models (Table S3)? What is the danger for the potential significance of the results of using fire data for a relatively short data period?

We slightly expanded on the limitations section of the discussion. In short, in the methods section, we justified the choice of 9 GCMs based on the availability of models for which at least three ensembles used daily minimum relative humidity and maximum temperature in the calculations. There would be a higher risk of errors using single ensemble members, hence our decision to choose a more limited set of models that had the most robust sampling.

“Models with multiple ensemble members were used to aid in robustness. However, we are limited by only using output from nine climate models that had the necessary meteorological variables for calculating FWI and furthermore are unable to resolve finer-scale changes that might be pertinent to extreme fire weather or land-surface feedback given the use of non-downscaled climate model output.”

It is unclear what the danger would be in limiting our empirical analysis to the MODIS era 2002–2023. We simply use the year with the maximum forested burned area as a reference case in our systematic quantification of contributing climate factors. We do not attempt to quantify whether such years are representative of 1-in-X year extremes from the perspective of burned area using GEV. It is likely that there would be much wider uncertainty in undertaking such an analysis and hence we make no statements on the unusual nature of the most extreme fire year in the 2002–2023 period and consequently do not discuss any potential significance of inferring return periods from strictly the fire database.

SPECIFIC COMMENTS

Line 1: Risk and Odds ratio are not the same. However, the authors treat them as the same thing.

We do not refer to “odds ratio” in the manuscript. We do refer to “odds” as is probability in the title and elsewhere, but do not think this is problematic as it aids in interpretability for those not familiar with risk ratio.

Lines 69-70. This sentence is not true. There are papers defining extreme wildfire events. Please see, for example, Tedim, F., Leone, V., Amraoui, M., Bouillon, C., Coughlan, M. R., Delogu, G. M., ... & Xanthopoulos, G. (2018). Defining extreme wildfire events: Difficulties, challenges, and impacts. *Fire*, 1(1), 9.

The point here is that while papers have discussed extreme wildfires, the definitions vary substantially, leading to a lack of an agreed-upon working definition of extremes. The paper referenced here mentions many of the challenges in working definitions, which is the point we are trying to highlight. We have cited this paper here to further this point.

Line 244: Please, include a space or non-breaking space between 500 and “m”. Please check all other units.

Yes, thanks for catching this. We have made the change.

Line 352: The risk ratio is not defined. A clear definition of risk ratio is needed.

We have now added a brief definition of risk ratio in the main text where it is first referenced.

“The RR is calculated as the ratio of the probability of FWI extremes exceeding the benchmark 1-in-15-year value under the contemporary climate to the counterfactual (here quasi-preindustrial) climate.”

Tables & Figures

The data and results presentation must be more rigorous. The Tables and Figures must be self-explanatory, and the Figures of this manuscript are not.

We have addressed these concerns by making numerous changes to the figures and tables in the interest of them being self-contained and interpretable. Thank you for highlighting these items.

Table 1

The reader may interpret MODIS as the burned area, GFA as the number of fires and also the number of “very large” fires, and GFED4s as the carbon emissions. The authors should clarify the relationship between the variables and the acronyms, otherwise, it seems to be an unnecessary and confusing definition of acronyms.

The “number of fires $\geq 100 \text{ km}^2$ ” should be “number of fires with burned area greater or equal to 100 km^2 ” or something similar.

Unnecessary and confusing definition of acronyms. Why define acronyms (e.g., GFA) not used in the Table? What is the relationship between “the number of fires” and “GFA”? Why define/mention GFA for both the number of fires and the number of fires with $\text{BA} \geq 100 \text{ km}^2$?

Anomalies are defined differently. In the title, “Anomalies” are “% of average for non-extreme years”, while in the caption, “Anomalies are compared concerning the mean calculated from all remaining years ...”. These definitions may or may not be the same. Authors must clarify what an extreme year is. How many extreme years are defined?

The use of * and + is confusing. These characters should be in superscript, otherwise $100+$ can be more than 100.

We have made the following changes (i) fully spelled out the number of large fires, (ii) made * and + superscript, (iii) removed the data sources from the caption, and (iv) reiterated our definition of extreme fire years as the year with the highest burn area during the MODIS period (2002-2023). We feel that the anomalies are adequately addressed in the caption as the percent departure from the averages calculated from all non-extreme years.

Table S1

Why is this table needed? Why don't you merge the information from Table 1 and Table S1?

Why use a different citation style in the Table line for Chile?

We have now consolidated Table 1 and Table S1.

Figures

Figure 1

Do not use the minus sign to define period; use n-dash. Please check and correct the entire manuscript.

We have implemented this suggested change throughout.

A title in each panel will increase the readability of the Figure

Thank you for the suggestion. We have now added a title for each panel in Figure 1.

Panel c, why circles of different sizes? What is the meaning of different sizes?

The year with the maximum ranking had a larger size than the others. We have now eliminated this visual to avoid confusion.

It seems like there is an unfinished sentence “Quantitative ranking of three FWI metrics (FWI_{fs}: max 90d moving mean FWI during the year; FWL₉₅: # of days of 95th percentile FWI, FWI_{max}: maximum daily FWI in the year) coincident to”

The sentence continued hereafter. We rephrase this to improve readability as:

“(c) Quantitative ranking relative 1979–2023 of three Fire Weather Index (FWI) metrics (FWI_{fs}: max 90d moving mean FWI during the year; FWL₉₅: # of days of 95th percentile FWI, FWI_{max}: maximum daily FWI in the year) coincident to, and ET_{yr-1} evapotranspiration from the prior year, corresponding to the year with the greatest burned area during 2002–2023.”

All acronyms must be defined, including FWI.

These are now defined.

Figure 2

The title of this Figure does not seem suitable. The Figure only presents estimates of the return periods of extreme FWI metrics.

The titles of each panel do not seem suitable either. It seems that you are presenting FWI_{fs}, FWI_{95d} and FWI_{max}.

All acronyms must be defined.

Figure 1 and Figure describe differently the areas in grey. Please harmonize.

- We feel that the “title” in the caption “**Extreme regional fire years co-occur with extreme fire weather.**” adequately synthesizes the results from the figure. This is unchanged.
- We added brief titles for the subpanels to assist readers in interpretation. The full descriptions of the metrics are now spelled out in the caption and all acronyms are defined.
- We have harmonized all maps to ensure the areas are greyed consistently.

Figure 3

The titles of each panel do not seem suitable either. It seems that you are presenting the fire characteristics and not the anomalies of FWI_{fs}, FWI_{95d} and FWI_{max}.

We believe the reviewer is referring to the wrong figure (likely Figure 2 of Figure 5). Figure 3 panels are (a) number of fires, (b) number of very large fires, and (c) fire carbon emissions. No change made.

Figure 4

How are the Box plots defined? Did you use P90, P75, P50, Mean, P25, P75, P10, variance, IQR?

The boxplots show the IQR with the horizontal black line denoting the median. We added this detail to the figure caption now.

Figure 5

As in Figure 3, the titles of each panel do not seem suitable either. It seems that you are presenting the fire characteristics and not the median risk ratio of FWI_fs, FWI_95d and FWI_max.

We believe the reviewer is referring to the wrong figure (Figure 3).

Please check and correct the figures in the supplementary material based on previous comments and suggestions.

We have implemented such changes to the supplement so that the figures are self-contained.